# The Role of Aberrant DNA Methylation in Misregulation of Gene Expression in Gonadotroph Nonfunctioning Pituitary Tumors

**DOI:** 10.3390/cancers11111650

**Published:** 2019-10-25

**Authors:** Paulina Kober, Joanna Boresowicz, Natalia Rusetska, Maria Maksymowicz, Agnieszka Paziewska, Michalina Dąbrowska, Jacek Kunicki, Wiesław Bonicki, Jerzy Ostrowski, Janusz A. Siedlecki, Mateusz Bujko

**Affiliations:** 1Department of Molecular and Translational Oncology, Maria Skłodowska-Curie Institute—Oncology Center, 02-034 Warsaw, Poland; paulina.kober@coi.pl (P.K.); joanna.boresowicz@gmail.com (J.B.); natarusetska@gmail.com (N.R.); jas@coi.waw.pl (J.A.S.); 2Department of Pathology and Laboratory Diagnostics, Maria Skłodowska-Curie Institute—Oncology Center, 02-034 Warsaw, Poland; maria.maksymowicz@coi.pl; 3Department of Genetics, Maria Skłodowska-Curie Institute—Oncology Center, 02-034 Warsaw, Poland; agapaziewska@poczta.onet.pl (A.P.); michalina.dabrowska@coi.pl (M.D.); jostrow@warman.com.pl (J.O.); 4Department of Gastroenterology, Hepatology and Clinical Oncology, Medical Center for Postgraduate Education, 01-813 Warsaw, Poland; 5Department of Neurosurgery, Maria Skłodowska-Curie Institute—Oncology Center, 02-034 Warsaw, Poland; jacek.kunicki@coi.pl (J.K.); neurochirurgia@coi.waw.pl (W.B.)

**Keywords:** DNA methylation, epigenetics, pituitary, adenoma, NFPA, gonadotrophinoma, gene expression

## Abstract

Gonadotroph nonfunctioning pituitary adenomas (NFPAs) are common intracranial tumors, but the role of aberrant epigenetic regulation in their development remains poorly understood. In this study, we investigated the effect of impaired CpG methylation in NFPAs. We determined DNA methylation and transcriptomic profiles in 32 NFPAs and normal pituitary sections using methylation arrays and sequencing, respectively. Ten percent of differentially methylated CpGs were correlated with gene expression, and the affected genes are involved in a variety of tumorigenesis-related pathways. Different proportions of gene body and promoter region localization were observed in CpGs with negative and positive correlations between methylation and gene expression, and different proportions of CpGs were located in ‘open sea’ and ‘shelf/shore’ regions. The expression of ~8% of genes differentially expressed in NFPAs was related to aberrant methylation. Methylation levels of seven CpGs located in the regulatory regions of *FAM163A*, *HIF3A* and *PRSS8* were determined by pyrosequencing, and gene expression was measured by qRT-PCR and immunohistochemistry in 83 independent NFPAs. The results clearly confirmed the negative correlation between methylation and gene expression for these genes. By identifying which aberrantly methylated CpGs affect gene expression in gonadotrophinomas, our data confirm the role of aberrant methylation in pathogenesis of gonadotroph NFPAs.

## 1. Introduction

Pituitary adenomas are frequently diagnosed intracranial tumors. These tumors may originate from various types of functional pituitary cells, and they exhibit specific endocrinological symptoms in patients resulting from hypersecretion of particular pituitary hormones. A large group of pituitary tumors, the so-called nonfunctioning pituitary adenomas (NFPAs), develops without endocrinological syndromes [1]. The vast majority of NFPAs originate from gonadotroph pituitary cells and are diagnosed by immunostaining for the transcription factor SF1 as well as FSH, LH, and α-subunit.

The incidence of sporadic point mutations in NFPAs is low [2], whereas changes in DNA methylation patterns and gene expression [3,4] are frequently observed in these tumors. Hence, impaired epigenetic regulation and changes in the pattern of epigenetic modification are thought to contribute to the development of gonadotroph pituitary tumors.

DNA methylation plays a role in complex, multi-factor epigenetic regulation of gene expression [5]. According to a generally accepted model, DNA methylation at gene regulatory sites suppresses transcription. Alterations in the methylation pattern, including both addition and removal of methyl groups, affect the cell’s gene expression profile and may facilitate the acquisition of tumorigenic potential [5].

Several previous studies performed genome-wide methylation profiling of pituitary adenomas. Some of these reports focused mainly on comparing hormone-secreting vs. non-functioning adenomas. The results revealed that NFPAs are the PA subtype that is most affected by aberrant DNA methylation [6,7,8]. Another study was focused solely on NFPAs and reported a difference in DNA methylation patterns between tumors exhibiting invasive vs. non-invasive growth, a feature of particular clinical importance [9].

Our previous comparison of the DNA methylation profile of NFPAs and normal pituitary samples indicated that abnormal methylation is involved in tumor development [10]. However, it remains to be elucidated how aberrant DNA methylation affects the profile of gene expression in NFPAs.

In this study, we applied high-throughput expression profiling to determine the gene expression pattern in the same tissue samples previously used for DNA methylation profiling. Our goal was to investigate in detail the role of DNA methylation in the misregulation of gene expression in gonadotroph NFPAs.

## 2. Results

### 2.1. Aberrant DNA Methylation in Gene-Associated CpGs in Gonadotroph NFPAs

After probes with missing intensity signals, probes located on sex chromosomes, and probes aligned either to multiple locations or to SNPs were filtered out, 407,939 probes on the HumanMethylation450K arrays (Illumina) remained for inclusion in the analysis. Comparison of DNA methylation profiles between NFPAs and normal pituitary samples revealed 23022 differentially methylated CpG positions (differentially methylated probes, DMPs; delta β value > 0.2 or < −0.2; adjusted *p* > 0.005) located within or near known human genes, according to the annotation provided by Illumina (Appendix A). These DMPs are annotated to 8594 human genes. The majority of these CpGs were hypermethylated in NFPAs (19658 CpGs, 85.4%), and the remainder (3364, 14.6%) are hypomethylated (Figure 1A). Both hypermethylated and hypomethylated methylation sites occurred in 5’ promoter regions (including TSS1500, TSS200, 5’UTR, and first exon) and gene bodies (including 3’UTR) with comparable frequencies (Figure 1B). When probes were stratified according to CpG content, hypomethylated DMPs include a notably higher percentage of CpGs located in open sea (67% vs. 51%) and a lower percentage of CpGs located in shelf and shore regions (24% vs. 37%) (Figure 1B). DMPs are listed in Appendix A.

### 2.2. Gene Expression Analysis for CpGs Differentially Methylated in NFPAs and Normal Pituitary Tissue

NFPA samples for which DNA methylation profiles had been determined were subjected to NGS-based measurement of gene expression using amplicon-based library preparation. Sequencing of Ion AmpliSeq Transcriptome libraries generated an average of 10,326,892 reads per sample, which were mapped to hg19 AmpliSeq Transcriptome version 1. After low-expression mRNAs were filtered out, 12778 mapped genes (approx. 61% of the Ion AmpliSeq Transcriptome) remained for inclusion in subsequent analyses.

We used Spearman correlation analysis to identify DMPs for which the DNA methylation level was related to the expression level of the associated gene. A significant correlation (*p* < 0.05) was observed for 2523 DMPs (11% of all gene-related DMPs) that are annotated to 1470 genes (17% of all the gene-annotated differentially methylated CpGs), as listed in Appendix A.

For the majority of these CpGs (1756 DMPs), methylation and gene expression were negatively correlated (median Spearman R coefficient of −0.463 ranging from −0.368 to −0.886). This negative correlation was observed both for CpGs located near gene promoters (regions ranging from −1500 bp from the transcription start site (TSS) to the first exon: HM450 probes classified as TSS1500, TSS200, 5’, or first exon, according to Illumina’s annotation) and for those located near gene bodies (gene body and 3’ HM450 probes, according to Illumina’s annotation) (Figure 2A). On the other hand, CpG methylation and gene expression levels were positively correlated for 767 DMPs (median Spearman R coefficient of 0.448 ranging from 0.368 to 0.845). In contrast to the negatively correlated DMPs, the positively correlated DMPs were mainly located in gene bodies (Figure 2A): The ratio of CpGs in gene bodies vs. promoters was 1.36 for negatively correlated DMPs and 2.5 for positively correlated DMPs (Chi-square test *p* < 0.00001).

When we classified CpG sites according to CG content, a similar proportion of DMPs located at CpG islands was observed among DMPs with negative and positive correlation. On the other hand, the proportions of shelf/shore CpGs differed between negatively and positively correlated DMPs, and were 45% and 28%, respectively. The proportions of open sea CpGs also differed: They represented 44% of negatively correlated, and 59% of positively correlated DMPs (Figure 2A). A detailed list of DMPs with methylation levels that were correlated with the expression of the associated genes is presented in Appendix A.

Subsequently, we performed gene set enrichment (GSE) analysis for 1470 genes whose expression was correlated with CpGs that were differentially methylated in NFPAs. The results revealed significant enrichment in multiple Gene Ontology (GO) Biological Process terms and Kyoto Encyclopedia of Genes and Genomes (KEGG) pathways that play well-documented roles in tumorigenesis. This include GO biological processes terms: Regulation of cell migration (GO:0030334), regulation of small GTPase mediated signal transduction (GO:0051056), transmembrane receptor protein tyrosine kinase signaling pathway (GO:0007169), negative regulation of transcription, DNA-templated (GO:0045892) among top processes as well as KEGG terms: Rap1 signaling pathway (hsa04015), Focal adhesion (hsa04510), Pathways in cancer (hsa05200), MAPK signaling pathway (hsa04010) among top identified pathways. The top ten most significantly enriched GO and KEGG terms are shown in Figure 2B, and all enriched pathways are listed in Appendix A.

### 2.3. DNA Methylation of Genes with Differential Expression Level in NFPAs and Pituitary

Comparison of gene expression profiles between NFPAs and normal pituitary sections identified 3758 differentially expressed genes (DEGs), listed in Appendix A. We focused on DEGs associated with DMPs for which DNA methylation and expression were correlated. This set of genes includes 444 of 1470 genes with a methylation/expression correlation, as shown in Figure 3A and listed in detail in Appendix A. These 444 genes are associated with 857 DMPs. The vast majority of these CpGs were hypermethylated in NFPAs (778/857, 90.1%), whereas only a minor fraction was hypomethylated (79/857, 9.2%).

Next, we sought to determine whether differential DNA methylation in NFPAs and the sign of the methylation/expression correlation corresponded to the direction of expression changes of particular genes. We anticipated that CpGs with negative methylation/expression correlations that were hypermethylated in NFPAs would be associated with genes that were downregulated in tumors. Indeed, most negatively correlated DMPs were hypermethylated in tumor tissue and located in genes that were downregulated in NFPA samples (532 CpGs, 238 genes) (Figure 3B). In addition, 42 DMPs hypomethylated in NFPAs associated with 33 overexpressed genes were identified among the negatively correlated CpGs. In turn, the most DMPs with positive DNA methylation/expression correlation were hypermethylated in NFPAs and associated with genes upregulated in adenomas (114 CpGs, 80 genes), whereas 23 hypomethylated DMPs were associated with 23 genes downregulated in NFPAs.

For the four categories of gene-annotated DMPs described above, the sign of the methylation/expression correlation in NFPA samples was concordant with the directions of the differences in DNA methylation and gene expression in NFPAs vs. normal pituitary. These DMPs were associated with 362 of 444 DEGs whose expression was correlated with aberrant CpG methylation level. For these genes, it is reasonable to conclude that tumor-related expression is related to impaired DNA methylation of particular CpG sites. GSEA of these 362 genes revealed significant enrichment in GO biological processes related to the regulation of RNA transcription (GO:0000122, GO:0006357), and response to laminar fluid shear stress (GO:0034616), but no enrichment in KEGG pathways. The top five GO processes are shown in Figure 3C.

Tumor-related changes in methylation levels and gene expression levels were discordant with the sign of methylation/expression correlation for 164 DMPs associated with 99 genes. These CpGs included DMPs with negative DNA methylation/expression correlations (i.e., genes that were hypermethylated but annotated to genes overexpressed in NFPAs, or hypomethylated but annotated to genes downregulated in NFPAs) and some with positive correlations (i.e., genes that were hypermethylated but downregulated in NFPAs or hypomethylated and overexpressed in NFPAs). For these 99 genes, the difference in gene expression in tumors vs. normal tissue was independent of DNA methylation of particular CpG sites, despite the fact that their expression and DNA methylation were correlated in tumor samples. The results are shown in Figure 3B where CpGs annotated to 362 genes, characterized by methylation difference and the direction of methylation/expression correlation is concordant with the direction of gene expression difference are presented in gray boxes.

### 2.4. DNA Methylation and the Expression of Selected Genes

To validate the results of our correlation-based DNA methylation/gene expression analysis of HM450 and NGS-based transcriptomic profiles, we determined the methylation levels of selected CpG sites, along with the expression levels of the associated genes, in samples of normal pituitary (n = 5) and tissue sections from an independent cohort of patients suffering from gonadotroph NFPA (n = 83). These patients were not included in the high-throughput methylation analysis or expression profiling.

We chose Cg26334801, located at *FAM163A*, as well as cg16672562 and cg05286653, located at *HIF3A*, because both of these genes were among the most downregulated in NFPAs (Figure 2B). Four CpGs located in the 5’ region of *PRSS8* gene (cg08775835, cg13439730, cg27436259, and cg03363863), whose methylation was correlated with *PRSS8* expression level, were also investigated. DNA methylation was assessed with pyrosequencing, and expression levels of *FAM163A, HIF3A*, and *PRSS8* were assessed with qRT-PCR. Forty-eight patients and five normal tissue sections were subjected to immunohistochemical staining in order to investigate protein expression.

Pyrosequencing confirmed that the DNA methylation level of particular CpGs was higher in NFPAs than in normal pituitary (Figure 4A). Downregulation of *FAM163A*, *HIF3A*, and *PRSS8* in pituitary gonadotroph tumors was clearly observed by both qRT-PCR and immunohistochemical staining (Figure 4B,C). Evaluation of immunostaining reactivity is presented in Table 1.

Next, using the NFPA samples from the validation group, we conducted a Spearman correlation analysis of DNA methylation data obtained for each CpG site and the expression level of the corresponding gene. The results confirmed the relationship between methylation and expression. A significant correlation of each CpG methylation with the expression level of the annotated gene was observed in the validation group of patients. The results are summarized in Table 2.

## 3. Discussion

Genome-wide DNA methylation patterns and methylation abnormalities in pituitary tumors have been reported previously [7,8,9,10,11]. To date, however, the impact of aberrant CpG methylation on gene expression in NFPAs has not been investigated by a multi-omics approach. In this study, we combined the results of methylome and transcriptome profiling of 33 gonadotroph NFPA samples to investigate whether DNA methylation at particular CpGs that are aberrantly methylated in gonadotroph tumors correlates with the expression levels of the corresponding genes.

Gonadotroph NFPAs exhibited an increase in DNA methylation relative to normal pituitary sections, and a large number of differentially methylated CpGs have been identified [10]. In our analysis, approximately 85% of CpGs that are differentially methylated in NFPAs and associated with known human genes were hypermethylated in tumors.

Our correlation-based analysis revealed that methylation of a major fraction of CpGs that are differentially methylated in gonadotroph adenomas and pituitary sections did not correlate with the expression of the annotated genes. This has also been observed in a similar analysis of other human cancers [12]. We observed a correlation between DNA methylation and gene expression in approximately 11% of gene-associated CpGs with aberrant DNA methylation.

These CpG sites are located in or near genes for which methylation and expression were previously reported to be correlated in pituitary adenomas, including *STAT5A* [10], *RHOD* [6], *GALNT9* [9], and *RASSF1* [13], as well as genes with previously described aberrant methylation in pituitary tumors, including *CDKN1A*, *TP73* [14], and *STAT3* [15]. A correlation between aberrant DNA methylation and the expression level was also observed for *HMGA2*, which plays an important role in the pathogenesis of pituitary tumors [16].

GSEA revealed that genes whose expression was related to the methylation level of differentially methylated CpGs were enriched in pathways and processes important for tumor development. For most CpGs whose methylation was related to gene expression, the relationship was an inverse one: For these sites, hypermethylation and hypomethylation corresponded to a decrease and increase in gene expression, respectively. These results are generally in line with the idea that in tumors, elevated methylation is associated with transcriptional silencing, providing a mechanism for inactivation of genes with tumor-suppressor function. On the other hand, hypomethylation may result in the activation of potential oncogenes [17].

To simplify the interpretation of the results, we distinguished two classes of aberrantly methylated CpGs: Those in promoters, i.e., located in the 5’ promoter gene region, which includes regions from 1500 bp upstream of the TSS up to the first exon of the gene, and those in gene bodies, including exons, introns, and 3’UTR sequences. For 81% of the CpGs located in promoters for which methylation and expression were correlated, the correlation was negative, in concordance with the overall model.

Approximately one-third of all methylation/expression-correlated CpGs were positively correlated, i.e., hypermethylation and hypomethylation were associated with increased and decreased expression, respectively. Among these CpGs, most are located in gene bodies, whereas only 22% are located in promoters, consistent with previously published reports. Gene body methylation has been observed in active genes [18], implying that it may have the opposite effect on gene expression to promoter methylation, i.e., increased DNA methylation in gene bodies may promote transcription [19].

In this study, we compared the expression profiles of NFPAs and normal pituitary sections without evidence of neoplastic features, obtained by Rathke’s cleft cyst surgery. This allowed us to identify DEGs and investigate which of these genes exhibits methylation-dependent expression. When we compared the list of genes with methylation/expression correlation and the list of DEGs, most genes were hypermethylated with a fold change in expression corresponding to the sign of the correlation, e.g., a gene with negative methylation/expression correlation that was hypermethylated and downregulated in tumors. In a subset of genes, however, the fold change in expression was inconsistent with the difference in DNA methylation and the sign of the correlation, e.g., a hypermethylated gene with negative methylation/expression correlation that was upregulated in tumors. These genes were excluded from the functional GSEA. We believe that the differential the expression of these genes in NFPAs and pituitary tissue was due to a mechanism distinct from aberrant DNA methylation, despite the fact that the expression levels of these genes were methylation-related. In general, our analysis showed that in gonadotroph NFPAs, the expression of 9% of DEGs is correlated with aberrant methylation.

To validate the results of our correlation-based analysis, we assessed DNA methylation of seven CpGs (from four particular genomic regions) that were hypermethylated in NFPAs and located within three genes that exhibited a methylation-related decrease in expression in tumors in an independent cohort of 90 patients. In the validation studies, we used alternative technical approaches, pyrosequencing, and qRT-PCR, to assess methylation and gene expression, respectively. In addition, we looked at genes that were not previously investigated in pituitary tumors, *FAM163A*, *HIF3A*, and *PRSS8*, which were hypermethylated in NFPAs. *FAM163A* and *HIF3A* were among the most downregulated genes in NFPAs. The function of *FAM163A* is unknown, but the gene is expressed in a highly tissue-specific manner in normal pituitary (according to https://gtexportal.org [20])*. HIF3A*, which encodes one of three subunits of hypoxia-inducible factor (HIF), is considered a negative regulator of HIF1A and a suppressor of hypoxia-inducible genes [21]. *PRSS8*, which encodes a serine protease that has been classified as a tumor suppressor, is downregulated in various cancers, including colorectal, liver, and esophageal cancer [22,23,24]. In esophageal tumors, *PRSS8* downregulation is related to DNA methylation [22].

The analysis of DNA methylation and expression confirmed the hypermethylation of selected CpG sites in NFPAs relative to normal pituitary, as well as a reduction in the expression of the corresponding genes at mRNA and protein levels. The independent validation group revealed a similar correlation of the seven selected CpG sites and the expression levels of the associated genes, providing validation for our large-scale correlation discovery analysis.

Our results provide insight regarding the role of aberrant DNA methylation in alterations of gene expression in NFPAs. In our correlation analysis, we identified genes whose expression was related to impaired DNA methylation, including about 350 genes that were differentially expressed in NFPAs and normal pituitary. It must be noted, however, that CpG methylation represents only a part of the range of epigenetic modifications. Gene expression is coordinated in multiple regulatory layers that include histone covalent modifications and the presence of histone variants, nucleosome remodeling, noncoding RNA as well as the expression of particular transcription factors.

Abnormal gene expression may result from mechanisms other than alterations in the DNA methylation pattern. For example, downregulation of *BMP4* in pituitary tumors is related to histone methylation, but is DNA methylation-independent [25], and *HMGA1* overexpression in pituitary adenomas is due to hypermethylation and silencing of the genes encoding for miRNAs targeting HMGA transcripts, rather than to direct methylation of the HMGA-encoding gene [26,27]. In general, predicting gene expression based on epigenetic data is challenging and requires data regarding different epigenetic mechanisms to be combined [28,29].

The key question is whether the epigenetic abnormalities observed in non-functioning pituitary tumors have clinical relevance. It is not clear whether DNA methylation changes play a role in the acquisition of clinically relevant biological features by tumors. Inconsistent results have been published, with some studies showing a clear relationship between methylation profile and tumor invasive growth [9] and other studies indicating only a very slight association [10]. However, it appears that in contrast to the small number of genetic alterations observed in NFPAs [2], the spectrum of methylation changes in these tumors is large, and aberrant methylation is observed in genes that generally play a role in tumorigenesis [30,31]. Therefore, DNA methylation abnormalities are thought to play a role in the pathogenesis of pituitary tumors [31,32]. We believe that our results support this idea by confirming the relationship between DNA methylation and gene expression profiles. Although methylation of only a subset of DMPs found in NFPAs is correlated with the expression of the corresponding genes, the genes with methylation-related expression are involved in processes and pathways relevant for neoplastic transformation.

## 4. Materials and Methods

### 4.1. Patients and Tissue Samples

The study enrolled patients with gonadotrophic clinically non-functioning adenomas (NFPAs) who underwent transsphenoidal surgery in Maria Skłodowska-Curie Memorial Cancer Center and Institute of Oncology in Warsaw, Poland. Following resection, tumor samples were snap-frozen in liquid nitrogen and stored at −80 °C. A fragment of each resected tumor was subjected to histopathological examination. WHO 2004 criteria were used for diagnosis [33], and tumor invasive growth was defined as described previously [10].

Gonadotrophic clinically non-functioning adenomas are the prevalent subtype of NFPAs. Genome-wide analyses of DNA methylation and gene expression were performed in 32 patients, and an additional 83 patients were subjected to PCR-based validation of the methylation and expression level of selected genes. The entire patient cohort included two NFPAs positive for both gonadotropins (FSH, LH, α-subunit) and TSH as well as eight samples negative for immunostaining with clear ultrastructural gonadotroph features. Patient characteristics are presented in Table 3. Detailed clinical data are presented in Appendix A. In addition to tumor samples, five sections of normal human pituitaries were used for DNA methylation profiling. Those reference samples were obtained from autopsies and underwent standard hematoxylin/eosin staining and subsequent histopathological evaluation to ensure the absence of incidental pituitary tumors. Autopsy samples were collected 25–46 h (median 35.5) after death. According to previous studies, this postmortem interval does not affect DNA methylation assessment by bisulfite-based methods [34] but may significantly affect the gene expression profile [35]. To address this problem, an additional five samples of formalin-fixed, paraffin-embedded (FFPE) tissue from histopathologically confirmed normal pituitaries obtained from resected Rathke’s cleft cyst were subjected to gene expression assessment by NGS and qRT-PCR and immunohistochemical analysis of protein expression analysis. A description of normal pituitary donors is presented in Appendix A. The Local Ethics Committee of Maria Skłodowska-Curie Memorial Cancer Center and Institute of Oncology in Warsaw approved experimentation on human patient samples (agreement 27/2017), and patients provided informed consent for the use of tumor samples for scientific purposes.

QIAamp DNA mini kit (Qiagen) was used for DNA isolation. DNA was stored at −20 °C prior to analyses. RNA was isolated using the RNeasy mini kit (Qiagen) after tissue homogenization with an Omni Tissue Master rotor-stator homogenizer (Omni International). The DNA and RNA quality was assessed spectrophotometrically using the NanoDrop 2000 (Thermo Scientific). Isolated RNA was stored at −80 °C prior to analysis.

### 4.2. Whole-Genome DNA Methylation Analysis

Genome-wide DNA methylation profile of 32 NFPA and five normal pituitary samples were determined using Infinium HumanMethylation450 BeadChips (HM450K) (Illumina, San Diego, CA, USA). Each sample was profiled in a single technical replicate. Data are deposited at Gene Expression Omnibus, GSE115783. Data were analyzed using the ChAMP data analysis pipeline [36] as described [10]. Probes with missing intensity signals and those with detection P-value above 0.01, along with probes located on sex chromosomes or aligning either to multiple locations or near SNPs, were discarded, as recommended [37]. Probes were annotated to genomic locations according to the lluminaHumanMethylation450k.db library. Differentially methylated CpGs were identified by comparing HM450K data from 32 NFPAs versus five normal pituitary samples. Delta β-value was used as measure of DNA methylation difference and calculated by subtracting the mean β-value of normal tissue from that of tumor samples. DMPs were defined as CpGs with delta β-value > 0.2 or < −0.2 and adjusted *p* > 0.005.

### 4.3. Gene Expression Analysis with Amplicon-Based Library Preparation and Next-Generation Sequencing (NGS)

Libraries for transcriptome sequencing were prepared using the Ion AmpliSeq transcriptome human gene expression kit (Thermo Fisher Scientific, Waltham, MA, USA). Multiplex PCR amplification of 100 ng of total RNA of each sample was performed using the Ion AmpliSeq transcriptome human gene expression core panel. Next, the amplified PCR products were subjected to enzymatic digestion followed by ligation of the oligonucleotide adapters. The samples were purified using Agencourt AMPure XP beads (Beckman Coulter, Brea, CA, USA) and stored at −20 °C for further processing. The length of DNA fragments and concentration of each library were assessed using high-sensitivity DNA analysis kits (Agilent, Santa Clara, CA, USA) on a 2100 bioanalyzer (Agilent). Each of the libraries was diluted to ~55 pM before template preparation. Up to seven barcoded libraries were subjected to template preparation using the Ion Chef Instrument and Ion PI Hi-Q Chef kit. Samples were sequenced on an Ion Proton instrument using PI chips. Sequencing reagents from the Ion PI Hi-Q sequencing 200 kit, provided as part of the Ion PI Hi-Q Chef kit (Thermo Fisher Scientific), were used for these experiments. Each sample was sequenced in a single technical replicate. Approximately 10 million reads were generated per sample. The raw unnormalized count matrix was generated from BAM files using the GenomicAlignments package [38] and imported to DESeq2 for data normalization and calculation of the estimates of dispersion [39]. Low-expression gene filtration was applied (genes with at least five sequencing reads in at least half of the samples were included), and subsequent analysis of differentially expressed genes between groups was performed by discrete distributions and negative binomial generalized linear models using DESeq2. Fold change of expression (FC), calculated as the ratio of mean read counts in tumors vs. normal samples, was used as a measure of the expression difference between groups of samples. Differentially expressed genes (DEGs) were defined as those with adjusted *p*-value < 0.05 and FC > 2 or FC < 0.5. Data have been deposited in the Gene Expression Omnibus database under accession number GSE136781.

### 4.4. Evaluation of DNA Methylation Pattern of Selected Gene Promoters

Pyrosequencing was used to measure DNA methylation levels of selected CpG sites. One microgram of each DNA was subjected to bisulfite conversion using the EpiTect kit (Qiagen, Venlo, Netherlands). PCR was performed in a 30-µL volume containing 1× PCR buffer, 2 mM MgCl_2_, 0.25 mM dNTPs, 0.2 μM each primer, and 0.5 U of FastStart DNA Polymerase (Roche Applied Science, Mannheim, Germany). The following cycling conditions were used: 94 °C for 3 min, 40 cycles of 30 s at 94 °C, 30 s at 55 °C, and 30 s at 72 °C, and a final elongation for 7 min at 72 °C. PCR amplicons were purified and analyzed using the PyroMark Q24 System (Qiagen) according to the manufacturer’s protocol. PCR primers are presented in Appendix A.

### 4.5. qRT-PCR Gene Expression Assessment

Expression levels of selected genes were assessed using qRT-PCR with SYBR Green-based detection. Reverse transcription of 500 ng of each RNA sample was performed using the transcriptor first strand cDNA synthesis kit (Roche Applied Science). Power SYBR Green PCR Master Mix (Thermo Fisher Scientific) was used for amplification. PCRs were run in 5-mL volumes containing 2.25 pmol of each primer on a 7900HT Fast Real-Time PCR System (Applied Biosystems, Foster City, CA, USA) in 384-well format.

Standard curves based on the amplification of known concentrations of cDNA template were used for determining PCR efficiency, and the 2^−ΔCT^ method was used for calculation of relative expression level. *GAPDH* was used as a reference gene based on previous validation [40]. PCR primer sequences are listed in Appendix A.

### 4.6. Immunohistochemistry

FFPE specimens derived from 20 patients with pituitary adenomas and six normal pituitaries were used for immunohistochemical staining. The procedure was performed on 4-μm tissue sections using the Envision Detection System (DAKO, Glostrup, Denmark). Sections were deparaffinized with xylene and decreasing concentrations of ethanol were used for rehydration. Samples were incubated in target retrieval solution pH 9 (DAKO) for 20 minutes in a 96 °C water bath. The retrieval solution was subsequently cooled for 25 minutes at room temperature, and the slides were treated with blocker of endogenous peroxidase (DAKO) for five minutes. Slides were incubated with polyclonal antibody (Ab) against HIF3A (MA5-26482, Thermo Scientific) (dilution 1:100, 60 minutes RT), FAM163A (PA5-52739, Thermo Fisher Scientific) (dilution 1:50 overnight 4 °C), or prostasin (PRSS8) (PA5-27977, Thermo Scientific) (dilution 1:200 60 minutes RT) and subsequently labeled with the Envision Detection System (DAKO). To obtain a colored reaction product, 3,3′-diaminobenzidine tetrahydrochloride (DAKO) was used as a substrate, and nuclear contrast was achieved by hematoxylin counterstaining. Results were recorded and assessed on a four-grade scale of staining intensity (0—none, +—weak, ++—moderate, +++—strong for staining) for each cell in a fixed field. The stained specimens were evaluated by a pathologist who was blinded to the qRT-PCR results.

### 4.7. Statistical Analysis

Quantitative continuous variables were analyzed by a two-sided Mann–Whitney U-test. The two-sided Fisher’s exact test was used to analyze proportions. A significance threshold level of α = 0.05 was adopted. Data were analyzed and visualized using GraphPad Prism (GraphPad Software, San Diego, CA, USA).

Correlation between DNA methylation and gene expression levels based on HM450K and NGS-based expression results was calculated using the Spearman method in the R environment. Methylation beta values normalized using the beta-mixture quantile (BMIQ) method, together with normalized read count values from the sequencing of amplicon-based mRNA libraries, were used. The EnrichR tool was used for gene set enrichment analysis (GSEA) [41]. GSEA terms were considered as significantly enriched when adjusted *p* < 0.05.

## 5. Conclusions

Approximately 10% of CpGs that are differentially methylated in gonadotroph NFPAs and normal pituitary are correlated with the expression of the associated genes. These genes are involved in various tumorigenesis-related pathways, and many of them are differentially expressed in NFPAs and normal pituitary. Validation of DNA methylation and the expression levels of selected genes confirmed the results from the correlation analysis performed on genome-wide data and showed that expression levels of *FAM163A*, *HIF3A*, and *PRSS8* were DNA methylation-related. In general, our results support the idea that aberrant DNA methylation plays a role in the pathogenesis of gonadotroph NFPAs.

## Figures and Tables

**Figure 1 cancers-11-01650-f001:**
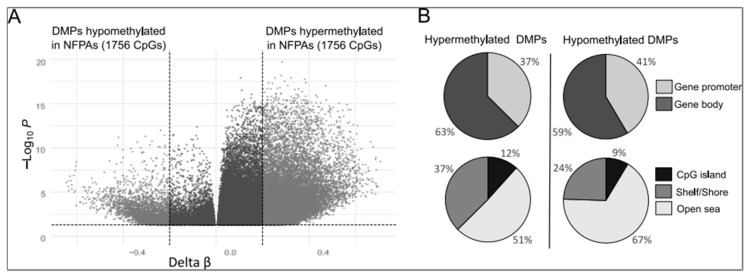
Distribution of gene-associated CpGs that were differentially methylated in gonadotroph nonfunctioning pituitary adenomas (NFPAs) and normal pituitary. (**A**) Volcano plots of differentially methylated CpG sites. (**B**) Differences in the proportions of aberrantly methylated CpGs, stratified according to gene position (5’ promoter including TSS1500, TSS200, 5’UTR, and first exon and gene body/3’ UTR), as well as CpG content (CpG island, open sea, and shelf/shore).

**Figure 2 cancers-11-01650-f002:**
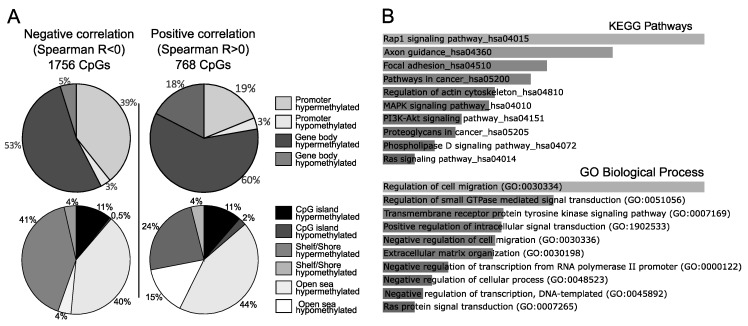
Results of correlation analysis between the methylation level of aberrantly methylated CpGs and the expression of the associated genes. (**A**) Distribution of aberrantly methylated CpGs for which DNA methylation was correlated with the expression of the associated gene in NFPA samples. (**B**) Top Kyoto Encyclopedia of Genes and Genomes (KEGG) pathways and Gene Ontology (GO) biological processes that are enriched for genes which expression is correlated with DNA methylation levels in NFPAs.

**Figure 3 cancers-11-01650-f003:**
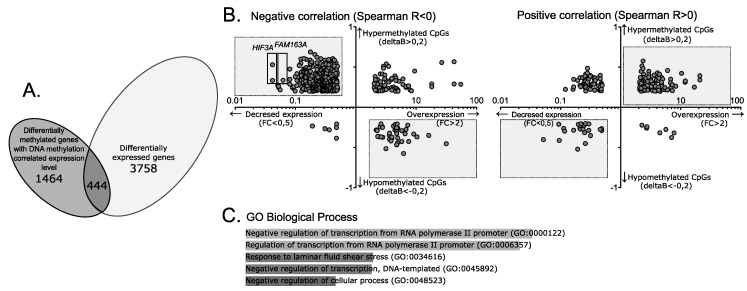
The role of CpG methylation in the expression of genes differentially expressed in gonadotroph NFPAs. (**A**) Overlap between genes for which expression level was correlated with aberrant CpG methylation and genes differentially expressed in gonadotroph NFPAs and normal pituitary sections. (**B**) Analysis of the direction of the DNA methylation difference and the direction of expression change of genes associated with CpGs with negative and positive methylation/expression correlations. Each dot represents a particular CpG site. The difference in the DNA methylation level, where delta β > 0 denotes hypermethylation and delta β < 0 denotes hypomethylation, is presented on the y-axis. Fold change in gene expression is presented on the x-axis. CpGs with a methylation difference and a sign of methylation/expression correlation concordant with the direction of the gene expression change are shown in gray boxes. (**C**) The top five Gene Ontology (GO) processes enriched for differentially expressed genes with expression levels correlated with methylation levels of the annotated differentially methylated CpGs.

**Figure 4 cancers-11-01650-f004:**
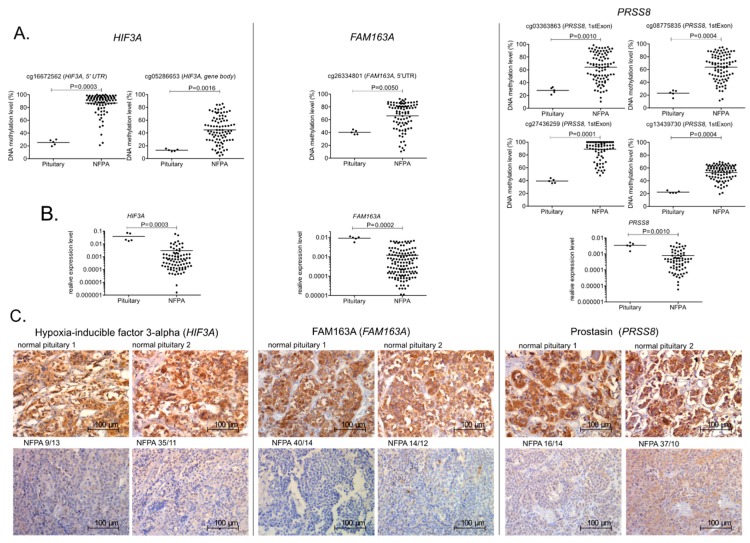
Difference in DNA methylation at regulatory regions, and expression levels of *HIF3A*, *FAM163A*, and *PRSS8* in gonadotroph NFPA samples and normal pituitary. (**A**) Pyrosequencing analysis of CpG sites. Each dot represents the methylation level in a particular sample. Mean values are shown as horizontal lines. (**B**) Relative expression levels of *HIF3A*, *FAM163A*, and *PRSS8* in normal pituitary and NFPAs. Each dot represents the expression level in the sample. Mean values are shown as horizontal lines. (**C**) Examples of immunohistochemical staining of normal pituitary and NFPAs with antibodies against HIF3A, FAM163A, and prostasin (PRSS8). Magnification, ×400.

**Table 1 cancers-11-01650-t001:** Results of immunostaining intensity in gonadotroph NFPAs and normal pituitary.

	*HIF3A*	*FAM163A*	*PRSS8*
NFPA			
Staining intensity	Number of samples (percentage)	Number of samples (percentage)	Number of samples (percentage)
strong (+++)	0/42 (0%)	0/42 (0%)	4/42 (9.5%)
moderate (++)	13/42 (30.9%)	11/42 (26.2%)	14/42 (33.3%)
weak (+)	25/42 (59.5)	25/42 (59.5%)	24/42 (57.1%)
0	4/42 (9.5%)	6/42 (14.3%)	0/42 (0%)
Normal pituitary			
Staining intensity			
strong (+++)	5/5 (100%)	5/5 (100%)	5/5 (100%)

**Table 2 cancers-11-01650-t002:** Comparison of the results of DNA methylation/gene expression correlation analysis from the study that used HM450K data and NGS-based transcriptomic profile (investigation group) and the validation study performed by pyrosequencing and qRT-PCR (validation group).

HM450K CpG Site	CpG Location	Gene	Investigation Group Spearman R; *p*-Value	Validation Group Spearman R; *p*-Value
cg26334801	5′UTR	*FAM163A*	−0.802; *p* < 0.0001	−0.688; *p* < 0.0001
cg16672562	5′UTR	*HIF3A*	−0.407; *p* = 0.0291	−0.380; *p* = 0.0005
cg05286653	Gene body	*HIF3A*	−0.490; *p* = 0.0076	−0.451; *p* < 0.0001
cg08775835	1stExon	*PRSS8*	−0.786; *p* < 0.0001	−0.487; *p* < 0.0001
cg13439730	1stExon	*PRSS8*	−0.559; *p* = 0,0016	−0.484; *p* < 0.0001
cg27436259	1stExon	*PRSS8*	−0.744; *p* < 0.0001	−0.568; *p* < 0.0001
cg03363863	1stExon	*PRSS8*	−0.765; *p* < 0.0001	−0.539; *p* < 0.0001

**Table 3 cancers-11-01650-t003:** Clinical characteristics of pituitary adenoma patients.

	Genome-Wide DNA Methylation/Whole Transcriptome Profiling	DNA Pyrosequencing/qRT-PCR
**NFPA patients (number of patients)**	32	83
Age (years)		
Range	36–85	34–82
Median	61	63
Gender (number of patients)		
Male	21	47
Female	11	36
Histopathology (number of patients)		
Gonadotroph PA	31	76
Null-cell/ Gonadotroph PA*	1	7
Clinical classification (number of patients)		
Invasive NFPA	17	50
Non-invasive NFPA	15	26
Unknown	-	7

* null cell adenomas with clear ultrastructural gonadotroph features.

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
