# Peer review of "The Role of Aberrant DNA Methylation in Misregulation of Gene Expression in Gonadotroph Nonfunctioning Pituitary Tumors"

_cancers, 2019, doi:10.3390/cancers11111650_

Round 1
Reviewer 1 Report
The manuscript submitted by Mateusz Bujko and co-workers present a detail analysis of CpGs methylation profile in gonadotroph NFPAs in reference to gene expression. The authors confirm previous genome-wide analysis based on correlation between the methylation/expression, showing reverse correlation in tumors between analysed parameter (an increase of methylation is associated with transcriptional silencing and hypomethylation may results in activation of gene expression).
The introduction is coherent and well brings into the topic of the paper. The methods for validation the hypothesis are properly chosen and described in sufficient details. The studies performed, are properly designed and controlled, replicated suitable. The presented data are of high quality (some comments to be addressed are below). Statistical analysis of data is performed properly. The conclusions reached, are consistent with the presented data.
- Figure 2B, presenting KEGG pathways and GO Biological Process, is not clear enough to read as the size of the font is too small. Also the description of the Figure 2B (line 116-126) seems to directly repeat the information included on the chart.
- Figure 3C is illegible.
- Some minor misspelling should be removed, e.g. lines: 75, 135, 226, 271, 281, 282, 289…
The presented study gives a valuable input into the field of epigenomic/transcriptomic analysis of tumor biology. The paper is very interesting and worth publishing.
Author Response
The authors thanks the Reviewer for the evaluation and commenting on the manuscript. Some amendments has been introduced in the manuscript and the text underwent extensive English correction. All the changes in the manuscript were introduced in Track Changes mode.
Please find the replies to the comments below.
Reviewer’s comment 1: Figure 2B, presenting KEGG pathways and GO Biological Process, is not clear enough to read as the size of the font is too small. Also the description of the Figure 2B (line 116-126) seems to directly repeat the information included on the chart.
Reply: Font size has been corrected. Our intention was to mention few the most important pathways in the text and to list all top 10 pathways on the figure. This has been corrected in the revised manuscript.
Reviewer’s comment 2: Figure 3C is illegible.
Reply: This figure has been corrected by changing font size.
Reviewer’s comment 3: Some minor misspelling should be removed,
Reply: Professional proofreading by Bioedit.com service was applied
Reviewer 2 Report
In this manuscript, Kober et al. studied the aberrant DNA methylation as well as transcriptomic profiles in gonadotroph nonfunctioning pituitary adenomas (NFPAs). By correlation analyses, they determined the ratio of differentially expressed genes that may be regulated by DNA methylation. They further analyzed the tumor-related pathways that may be affected by DNA methylation. Finally, through pyrosequencing (for DNA methylation) and qPCR/IHC (for expression levels), the authors confirmed the correlation of aberrant DNA methylation and altered gene expression of FAM163A, HIF3A and PRSS8 in NFPAs.
Whole genome methylation analysis of NFPA has not been well studied. So the manuscript provide some useful information for NFPA researchers. However, the clinical impact of DNA methylation in NFPA remains to be an open question. Since NFPA is a low grade tumor, it is not sure whether DNA methylation has clinical values. The authors may want to discuss this point in the manuscript. Meanwhile, the authors may want to improve the manuscript by providing more data and technical details.
Main concern:
The reproducibility of high through-put sequencing is a common problem. Because this manuscript is mainly an analysis and validation of the sequencing data and lacks functional assays with regard to certain genes or pathways, it is critical that the sequencing data are reproducible. The authors may want describe the technical details for the sequencing data. How many repeats did they perform? Did they make cross comparison of 32 sequencing results to get the commonly hyper-/hypo-methylated genes? The details of gene expression analysis should also be provided. Although they performed some correlation analyses and tried to confirm the regulation of gene expression by DNA methylation in NFPAs, the manuscript lacks direct evidence. Given that it may be difficult to get NFPA cells for cell culture, the reviewer suggested that the authors may be able to treat fresh NFPA tissue slides with demethylation drugs such as 5-aza in slide culture, and then determine the upregulation of gene expression after demethylation. IHC staining images were not clear and lacked scale bars. Language may be improved by consulting a native speaker.Author Response
The authors thanks the Reviewer for the evaluation and commenting on the manuscript. Some amendments has been introduced in the manuscript and the text underwent extensive English correction. All the changes in the manuscript were introduced in Track Changes mode.
Please find the replies to the comments below.
Reviewer’s comment 1: Whole genome methylation analysis of NFPA has not been well studied. So the manuscript provide some useful information for NFPA researchers. However, the clinical impact of DNA methylation in NFPA remains to be an open question. Since NFPA is a low grade tumor, it is not sure whether DNA methylation has clinical values. The authors may want to discuss this point in the manuscript.
Reply: We agree that clinical significance of DNA methylation changes in NFPAs is not clear. As pointed by the reviewer, NFPAs are slow growing tumors and patients generally have good prognosis. This hampers the evaluation of clinical significance of methylation changes. In general, patients outcomes are rather more related to the quality of tumor resection than tumor’s biological features. The role of DNA methylation in invasive growth of NFPAs as the most important clinical feature has been evaluated previously, but the reports are inconsistent. For example, in some of the studies a clear relationship between methylation profile and tumor invasive growth was found [1] while this relationship was very slight in other results [2].
In our study we made an attempt to determine whether DNA methylation abnormalities may contribute to pathogenesis of non-functioning pituitary tumors. It is difficult to address this issue with functional investigation since there are no cell line models of human NFPAs and nearly no methods for manipulation of methylation level of particular region in vitro. Therefore, functional analyses that are normally used for investigating the role of driver genomic mutations are not available for the research on the role of epigenetic abnormalities.
Despite the fact that direct evidence is not available, we believe that our results support the idea that DNA methylation abnormalities play a role in the development of NFPAs:
It appears that the large number of DNA methylation changes is present in NFPAs, in contrary to the low level of genomic alterations found in these tumors.
Some DNA methylation changes affect the expression of particular genes, including genes that are involved in key cancer pathways.
As suggested by the Reviewer, this issue was discussed in the revised manuscript in the end of Discussion section.
Reviewer’s comment 2: The authors may want to improve the manuscript by providing more data and technical details.
Reply: Material and methods section of the revised manuscript was supplemented with more details in the description of the methods used.
Reviewer’s comment 3 The authors may want describe the technical details for the sequencing data.
Reply: We provided more details in the revised manuscript. We introduced the information on the number of probes from HM450K arrays filtered out for technical reasons and how many probes were included in the analysis as well as the information on how many reads were obtained in sequencing.
Reviewer’s comment 4 How many repeats did they perform?
Reply: Each of the sample was subjected to hybridization to methylation microarray and transcriptomic sequencing in single technical replicate. We aimed to include a possibly large number of samples for statistical evaluation (the cost of analysis was the limiting factor) not to multiply technical replicates.
Reviewer’s comment 5. Did they make cross comparison of 32 sequencing results to get the commonly hyper-/hypo-methylated genes?
Reply: The problem for cross comparison of our data with the results of the other studies is that very low number of datasets are available for cross comparisons. In public repository we found only 1 dataset that contains DNA methylation data for NFPAs (Gene Expression Omnibus GSE54415) from previously published study [3]. This dataset contains results of methylation profiling of 15 NFPA samples with HM450K (Illumina) arrays but does not include any data for normal pituitary to perform a comparison similar to that in our study. We looked whether we could use NFPA data from GSE54415 to compare to our data for normal pituitary samples, for cross comparison of two independent groups of NFPA samples. Unfortunately, the initial Multidimensional scaling analysis shows that NFPA samples from two datasets (GSE54415 and ours) differ much more than NFPAs and normal samples in our study, probably mainly for technical reasons (Figure 1). We are convinced that due to these differences, cross comparison of these two NFPA groups wouldn’t be informative.
Figure 1. The preliminary MDS analysis of two data sets of HM450K results showing differences between these two datasets implying their uselessness in cross comparisons of the results.
Reviewer’s comment 6. The details of gene expression analysis should also be provided.
Reply: We introduced more details in the description of the analysis of gene expression.
Reviewer’s comment 7. Although they performed some correlation analyses and tried to confirm the regulation of gene expression by DNA methylation in NFPAs, the manuscript lacks direct evidence. Given that it may be difficult to get NFPA cells for cell culture, the reviewer suggested that the authors may be able to treat fresh NFPA tissue slides with demethylation drugs such as 5-aza in slide culture, and then determine the upregulation of gene expression after demethylation.
Reply: We agree that 5-aza cytidine is an inhibitor of DNA methyltransferases that is commonly used when investigating the relationship between DNA methylation and expression level of particular gene. In most common experimental procedure the cell line is cultured in medium containing 5-aza for 3-5 days and subsequently the expression of the gene of interest is measured in 5-aza treated and untreated (control) cells. Unfortunately, no human NFPA cell lines are available, therefore this common approach is not applicable for investigating human gonadotropinomas.
Some authors used primary cultures of patients-derived NFPA cells but we don’t know any published study that utilized 5-aza treatment of primary cell culture derived from pituitary tumor. Since we have no experience and no established protocol for culturing the primary pituitary adenoma tissue cultures form patients we are not able to perform such potential experiment.
On the other hand, we’ve noticed another obstacle. 5-aza cytidine acts during cell divisions by inhibiting methyltransferases after DNA replication. Our own experience with 5-aza C treatment indicates that it works effectively in the highly proliferating cells like colon cancer cell lines HCT116 or HT29 [4, 5] but has a very low effect in the slowly proliferating cells like BenMen1 derived from benign meningioma (unpublished data). Therefore, we expect that it would be difficult to observe a notable effect of 5-aza cytidine in cells from pituitary adenoma which is generally slow proliferating benign tumor.
Its worth to note that according to available data it appears that Aza-C affects genes expression profile not only by changing DNA methylation pattern but also through many secondary mechanisms [6]. After 5-aza cytidine treatment a large number of genes can be misregulated in methylation-independent manner, therefore, the results of expression profiling (especially transcriptome-wide) after 5-aza treatment should be interpreted cautiously.
Reviewer’s comment 8. IHC staining images were not clear and lacked scale bars.
Reply: We increased the magnification of the pictures to make them more clear and added scale bars.
Reviewer’s comment 9. Language may be improved by consulting a native speaker.
Reply: Professional language editing by Bioedit.com service was applied.
References
Gu Y, Zhou X, Hu F, et al (2016) Differential DNA methylome profiling of nonfunctioning pituitary adenomas suggesting tumour invasion is correlated with cell adhesion. J Neurooncol 129:23–31. https://doi.org/10.1007/s11060-016-2139-4 Kober P, Boresowicz J, Rusetska N, et al (2018) DNA methylation profiling in nonfunctioning pituitary adenomas. Mol Cell Endocrinol 473:194–204. https://doi.org/10.1016/j.mce.2018.01.020 Ling C, Pease M, Shi L, et al (2014) A pilot genome-scale profiling of DNA methylation in sporadic pituitary macroadenomas: association with tumor invasion and histopathological subtype. PLoS One 9:e961178. https://doi.org/10.1371/journal.pone.0096178 Bujko Mateusz, Kober Paulina, Mikula Michał, Ligaj Marcin, Zwierzchowski Lech, Ostrowski Jerzy SJA (2014) Downregulation of PTPRH (SAP-1) in colorectal cancer. Clin Epigenetics Soc Meet Dusseld 2014 Bujko M, Kober P, Statkiewicz M, Mikula M, Ligaj M, Zwierzchowski L, Ostrowski J SJ (2015) Epigenetic-Mediated Downregulation of μ-Protocadherin in Colorectal Tumours. Gastroenterol Res Pr 2015:317093 Seelan RS, Mukhopadhyay P, Pisano MM, Greene RM (2018) Effects of 5-Aza-2′-deoxycytidine (decitabine) on gene expression. Drug Metab Rev 50:193–207. https://doi.org/10.1080/03602532.2018.1437446

Round 2
Reviewer 2 Report
The authors have addressed most of my concerns. There is only one point to be further clarified. Whereas it is acceptable that single technical replicate was performed, the authors didn't clarify whether the data were generated by comparison of all the sequencing data they got. They described the difficulty of cross comparison between their data and other lab's data, which was not my concern. The concern is how many sequencing datasets from their own lab were used to get the consistently changed methylation. Please clarify this point in the Methods section.
Author Response
Reviewer’s comment: The authors have addressed most of my concerns. There is only one point to be further clarified. Whereas it is acceptable that single technical replicate was performed, the authors didn't clarify whether the data were generated by comparison of all the sequencing data they got. They described the difficulty of cross comparison between their data and other lab's data, which was not my concern. The concern is how many sequencing datasets from their own lab were used to get the consistently changed methylation. Please clarify this point in the Methods section.
Reply: Sorry for the misunderstanding in 1st round of revision. In fact, we used all the available samples that has been analyzed for both DNA methylation (with HM450K arrays) and gene expression (with Ion AmpliSeq Transcriptome Human Gene Expression method).
For methylation analysis 32 tumor samples and 5 samples from normal pituitary were used. Data from methylation profiling of these samples are deposited as a single data set at gene expression omnibus. This deposited data set include HM450K data for 34 samples but we couldn’t get RNA for two samples for transcriptomic profiling, therefore these 2 sample were excluded from the analysis performed in this study. Thus, differentially methylated CpGs were identified by comparing HM450K data from 32 NFPAs versus 5 normal pituitary samples.
We modified methods section to clearly indicate that single technical replicate has been performed and the number of samples used for comparison and identification of differentially methylated sites. We hope it is clearly presented in the revised manuscript